# Methods for Nondestructive Testing of Urban Trees

**Richard Bruce Allison [1,2,\*], Xiping Wang [3,\*] and Christopher A. Senalik [3]**

1 Department of Forest and Wildlife Ecology, University of Wisconsin, Madison, WI 53706, USA
2 Allison Tree Care, LLC, Verona, WI 53593, USA
3 USDA Forest Service, Forest Products Laboratory, Madison, WI 53726-2398, USA; christopher.a.senalik@usda.gov
* Correspondence: rbruceallison@tds.net (R.B.A); xiping.wang@usda.gov (X.W.); Tel.: +1-608-848-2345 (R.B.A); +1-608-231-9461 (X.W.)

**Abstract:** Researchers have developed various methods and tools for nondestructively testing urban trees for decay and stability. A general review of these methods includes simple visual inspection, acoustic measuring devices, microdrills, pull testing, ground penetrating radar, x-ray scanning, remote sensing, electrical resistivity tomography and infra-red thermography. Along with these testing methods have come support literature to interpret the data.

**Keywords:** decay; defect; hazard assessment; inspection; nondestructive testing; urban trees

## 1. Introduction

Trees within an urban community provide significant ecological, economic and social benefits, making a city more livable and comfortable for its inhabitants [1]. However, as large physical wooden structures in close proximity to dense populations of people and property, tree failure can cause harm. Urban forest managers use biological and engineering principles to determine a tree's structural soundness and estimate the probability of failure. Nondestructive testing (NDT) methods by locating and quantifying wood decay and defect are used to measure the physical condition of trees within the urban forest to promote public safety and property protection. These methods are of special value to the urban forest managers and arborists responsible for the general safety of city residents, roadway transportation and utility corridors. Nondestructive testing methods used in the urban forest include visual examinations; acoustic methods of sounding, stress wave timers and multi-sensor tomography; microdrill resistance testing; ground penetrating radar; static and dynamic pull testing; CT X-ray; aerial remote sensing including drones [2–4]; electrical resistivity tomography (ERT); and infra-red thermography (IRT).

## 2. Visual

Visual inspection when combined with knowledge of tree biology and biomechanics can provide indications of a tree's internal wood condition and predictions of failure [5]. A tree's growth is responsive to its environment. It is a self-optimizing, mechanical structure. To allow uniform mechanical stress over its entire surface, additional wood is laid down over decayed or damaged areas. Thus, trunk bulges, wound wood, or extreme lower trunk flair at the root collar area can indicate concealed cavities, cracks or decay. Crown retrenchment, fungal conks or open cavities are further visual evidence of decay and structural defects. Visual inspections for tree stability evaluation are limited. Various nondestructive testing tools are available to urban forest managers to "see inside" trees, allowing them to make better decisions.

### 3. Acoustic

Mechanical sounding is a well-established forestry method of tapping a tree trunk with a wooden or rubber mallet and listening for the tell-tale drumming response indicating an interior hollow. This results from the sound wave attenuating or flattening as it progresses across a hollow transverse trunk section [6–8].

Acoustic stress wave timers are electronic portable tools that employ the principle that sound waves travel through wood at differing velocities depending upon density [9]. In addition, internal obstructions such as cracks or cavities will increase the perceived time of flight of impact-induced stress waves as they travel across the diameter of a tree trunk. Stress wave timers are used on standing trees in the urban forest as a simple nondestructive tool to determine the internal condition of the diameter path within the transverse section tested [10]. Stress wave timers consist of two accelerometers attached to nails, screws or spikes with a shallow wood connection just beyond the bark placed on opposite sides of the trunk or limb being sampled (Figure 1). The accelerometers are connected by electric cables to a main electronic circuit board designed to display, like a stopwatch, the time in microseconds for an impact-induced stress wave to travel from the start accelerometer to the receiving one across the trunk or limb wood. That number is then divided by the number of inches or centimeters in the diameter travel path to determine the time-of-flight per measurement units travelled. If one knows the expected time-of-flight of the species sampled, a comparison can be made to make a judgement regarding tree structural condition [11,12]. Figure 2 shows the field use of a Fakopp Microsecond Timer in testing a large maple tree on the University of Wisconsin Madison campus.

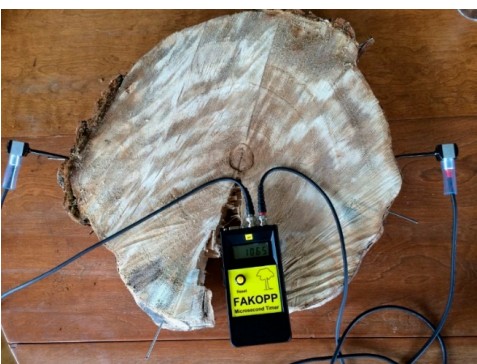

**Figure 1.** Concept of single-path acoustic method measuring time-of-flight on a tree disk.

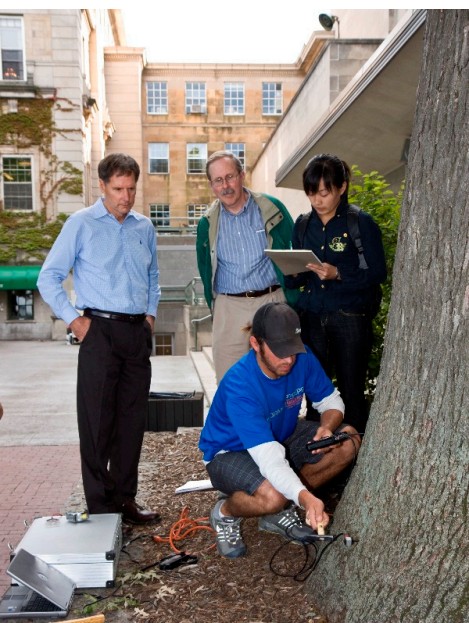

**Figure 2.** On-site tree inspection with a portable stress wave timer.

The use of stress waves to detect tree defects and decay is based on the observation that sound waves' movement through wood is directly related to the physical and mechanical properties of wood. Stress waves travel slower in decayed or deteriorated wood than sound wood. On hardwood trees such as oak and maple, one can expect a stress wave time of flight across the diameter to be from 18–22 μs/in. (7–9 μs/cm). Numbers higher than that indicate an obstruction interfering with the passage of the sound wave. This could be a crack, a cavity, or decay. On softwood trees with lower wood density, such as spruce, pine or hemlock, the stress wave will take a longer time, often producing numbers in the 25–30 μs/in. (10–12 μs/cm) range. The stress wave timer is a qualitative tool indicating either solid wood or a problem requiring further quantitative investigation to determine the location and extent of the problem.

## 4. Multi-Sensor Acoustic

Multi-sensor acoustic instruments are available [13] using the same principle as the single-path stress wave timer but employing as many as twelve accelerometers or more, spaced around the trunk circumference to create a matrix of measurements with results processed by computer projection software and displayed visually as a tomograph (Figure 3). Each of the accelerometers has a turn to serve as the sending one and the computer measures the time of flight to each of the other accelerometers. That information is processed by a field computer creating a computed tomography representation of the varying stress wave velocities across the trunk slice or transverse section measured. It is a self-calibrating scale showing multiple colors representing quadrants of perceived change in density from most dense to least. In interpreting test results, it is important to know that the tomograph picture is not a precise representation of the exact location and area of any internal defect but rather a display of the measured changes in sound velocity across the transverse section. A crack, for example, might cause a sound wave shadow larger than the actual defect. With careful interpretation, however, the acoustic tomographic tool can assist in gauging the extent, type and approximate location of the defect and the amount of remaining solid wood [14–18].

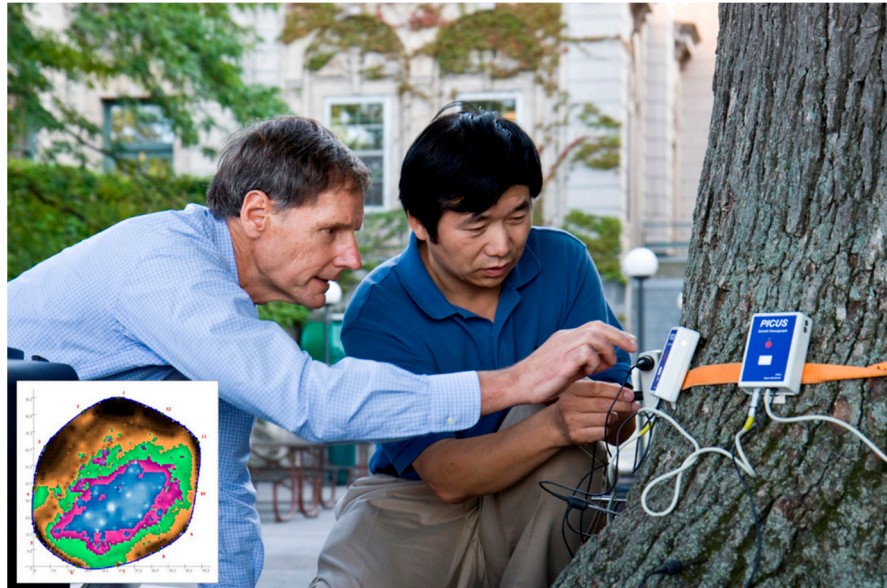

**Figure 3.** Multi-sensor acoustic tomography test to determine the extent and location of internal defects on a large tree.

## 5. Microdrill Resistance Tools

Microdrill resistance tools are a nondestructive quantitative method of displaying changes in wood density related to decay and cracks. Using a hand-held portable drill, a needle-like drill bit is projected into the trunk. Its torque and/or thrust resistance is measured as it passes through

the wood and is displayed on graph paper or a computer screen. Changes in amplitude on the graph reveal wood density variance and the size of internal decay pocket relative to the cross-section (Figure 4). The method and tool was developed by Frank Rinn as a graduate student at the University of Heidelberg [19] and further refined in subsequent years [20–22].

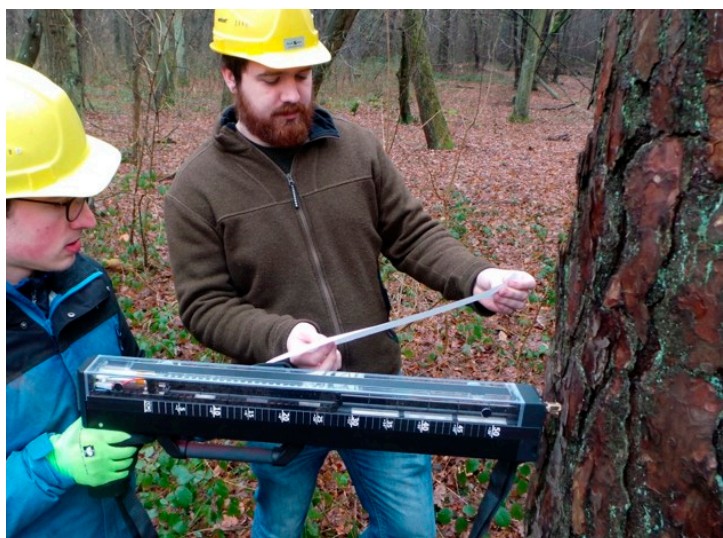

**Figure 4.** Using a Resistograph® tool to determine the internal condition of a tree (Photo credit: Frank Rinn).

## 6. Pull Testing

Gunter Sinn, German landscape architect, began working with Lothar Wessolly at the University of Stuttgart, Germany to investigate wind load simulation by pulling on urban tree trunks to create precisely measured static loads and movement response as a means to estimate structural stability and potential for uprooting [23]. The term Static Integrated Assessment (SIA) was used by Wessolly and Erb in their books [24,25] to describe a comprehensive method for predicting urban tree uprooting using static load tests. The tests require a rope or strap to be attached to the subject tree's upper crown attached to a winch with a dynameter or force meter measuring the amount of force exerted. An inclinometer records uprooting reaction and elastometers measure the elastic properties of the wood under load [26–29].

Methods of dynamic testing have also been researched, monitoring tree movement during wind events with inclinometers and elastometers but avoiding pulling on the tree trunk with ropes and winches [30,31]. The practicality and best methods for pull testing in the urban areas has generated debate. [32,33]

## 7. Ground Penetrating Radar

Radar was developed in the period around World War Two. The term Radar, first used in 1940, is an acronym for "radio detection and ranging." It uses radio waves to determine the range, angle or velocity of objects. A radar system consists of a transmitter producing electromagnetic waves, an antenna transmitting those waves, a receiving antenna and a processor to analyze the results. When the transmitted radar signals come in contact with an object, they are reflected back to the receiving antenna allowing the processor to determine the object's location in the space being observed. Radar's most common use in the urban forest is underground root detection. The transmitting antenna is pulled along the ground level with the radar signals reflected primarily off the moisture in roots allowing the processor to estimate the location of roots along a predetermined soil depth [34]. The method has been successfully employed by researchers investigating the distribution and infrastructure impact of roots within the urban forest [35,36]. Examinations of living tree trunks using ground penetrating radar

(GPR) has shown that accuracy is affected by tree diameter and regularity of shape. Small diameter and irregularly shaped trees adversely affected the accuracy of GPR [37]. Innovation in GPR ouput interpretation has resulted in the ability to distinguish between moisture pockets and voids [38,39]. Ongoing research is further refining the usage of GPR to find wood rot [40].

## 8. X-ray CT Scan

X-ray computed tomography (CT) scanning is a nondestructive testing method providing three-dimensional information about the internal inhomogeneous structure of sampled items. The CT method, developed by A.M. Cormack and G.N. Hounsfield in the 1970s, is now a standard testing method in medicine and material sciences [41]. It has a mathematical basis derived from the work of Johann Radon (1917) [42] who demonstrated that one can reconstitute the image of an object using a complete set of projections of relevant physical variables. CT, using ionizing radiation (x-ray or gamma ray), relies on the physical principle of absorption of high energetic photons passing through matter. A measurement of the lessening or attenuation of the energy source as it passes through a specimen is used to create a map of density variations of the internal inhomogeneous structure. Because medical scanners typically apply photon energy in the range of 25 to 150 keV, photoelectric absorption is the main cause of attenuation. The attenuation phenomena in wood is caused mainly by the Compton effect and is proportional to the wood's mass density, with density variations due to the distribution of anatomic structures and the water content in the cell walls and lumina. Arthur Holly Compton was awarded the 1927 Nobel Prize in physics for his observation that there is a decrease in energy (i.e., increase in wave length) of an x-ray or gamma ray as it interacts with matter.

The degree of energy attenuation is derived subtracting the number of transmitted photons arriving at the receiving sensor from the number generated at the initiation source. It is dependent on the mass attenuation coefficient, the thickness of the material, and the density.

The attenuation co-efficient is dependent upon the emitted energy spectra. That spectra is a function of the applied voltage, current and charge of the x-ray tubes within the scanner equipment. Therefore, absolute coefficients from different scans cannot be compared. It is necessary to normalize them against distilled water as an internal standard described in Equation (1).

$$\mu_{rel} = 1000 \times \frac{\mu_{material} - \mu_{water}}{\mu_{water} - \mu_{air}} \tag{1}$$

The relative attenuation coefficient, $\mu_{rel}$, is termed a Hounsfield Unit (HU) or sometimes a CT number. It is strictly correlated with bulk density. HU = 0 represents the density of water (1.0 g/cm$^3$) and HU = −1000 represents that of air. HU values greater than 0 represent materials with a bulk density greater than water.

CT images are obtained by the rotation of a radiation source and detectors around the specimen. Attenuation coefficients are converted into density data as displayed as images of the sample coded by color or a 256 unit gray scale. CT scanning has been shown to be an accurate measurement of wood density [43–45]. The technology is used for quality control of wood products and in measuring internal characteristics of sawmill logs to guide milling for maximum value.

A mobile CT apparatus was developed in Germany [46] that provided both research and urban forest hazard tree analysis functions. The drawback for its expanded use is that it used gamma ray with a source of radioactive isotope cesium and had a low spatial resolution. The development of mobile CT scan equipment using X-ray tubes as an energy source has applicability for field use by urban foresters [47].

## 9. Remote Sensing

Satellite imagery and aerial remote sensing technologies, particularly hyperspectral and lidar, are advancing in data gathering capabilities and resolution, creating additional opportunities for

nondestructive testing of the urban forest [48]. The increasing use of aerial drones plus software advances also present reductions in cost and ease of use in arboriculture [49,50].

## 10. Electric Resistivity Tomography and Infra-Red Thermography

Wood decay affects both wood moisture and electrolyte content properties in standing trees. Both properties can be measured by electric resistivity creating the opportunity with electrical resistivity tomography tools to determine the presence and extent of internal decay [51]. Damaged or abnormal tree tissue such as wood deterioration or void can also be detected based on the measured thermal difference using an infra-red camera [52]. Defects affect the internal energy flow leading to surface temperature differences.

## 11. Applying Information Gathered by Nondestructive Testing

Protocols for combined use of NDT tools and methods have been developed for efficient application in the urban forest. Nondestructive testing methods and tools provide extensive information on the internal inegrity of trees. The job of analyzing and applying this new information belongs to the urban foresters assigned the duty of safety management of these often massive tree structures in close proximity to building, roads and people. The International Society of Arboriculture has significantly advanced the science and practice of assessing tree stability and risk with the publication of "Tree Risk Manual" [53] and the development of a Tree Risk Assessment Qualification program to train and certify arborists in performing such assessments. The program comes with the caveat that it is impossible to eliminate all risk associated with trees. Nonetheless, the science of tree stability analysis makes an important contribution to not only public safety but also our enjoyment of trees by providing, if not a perfect, at least an improved method to measure tree stability, thus increasing our comfort level in the urban and landscape forest.

## 12. Concluding Remarks

The term non-destructive evaluation is a term well established in the wood science research literature [54–56], representing "the process by which selected physical properties of a material is being tested without damage or alteration to its end-use capabilities." It could be argued that micro-drill testing, acoustic testing, and electrical resistivity tomography create wounds in the bark or wood and that even tensioning with static/dynamic pull testing could stretch or rupture wood fibers. However, generally speaking, these and other techniques in this review leave the tested tree without significant damage or alteration to its end use and are generally recognized as non-destructive evaluating techniques in the urban forest. Of these techniques, visual inspection, single path and multi-sensor acoustics, micro-drill resistance tests and static/dynamic pulling tests are the most commonly used by arborists and urban forest managers. The equipment costs are within the practitioners' budget range, the science is well understood and there are multiple manufacturers and equipment distributors. The development and use of nondestructive test tools and methods have significantly improved the well-being of the urban forest and its inhabitants.

**Author Contributions:** R.B.A. conducted literature review and drafted the manuscript; X.W. reviewed and edited manuscript, provided visual images; C.A.S. reviewed and edited the manuscript. All authors have read and agreed to the published version of the manuscript.

**Funding:** This research received no external funding.

**Acknowledgments:** This review article is based on research collaborations between University of Wisconsin-Madison, Allison Tree Care, LLC, and the USDA Forest Service Forest Products Laboratory.

**Conflicts of Interest:** The authors declare no conflict of interest.

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
