# Peer review of "Methods for Nondestructive Testing of Urban Trees"

_forests, doi:10.3390/f11121341_

Round 1
Reviewer 1 Report
The topic of this paper is still relevant. Literature in the field presents numerous studies in which various tree testing methods have been investigated with the purpose of determining tree stability and the risk that trees pose. However, the article does not have a high degree of novelty and originality. There are numerous articles that have approached this topic over the past 30 years. The methods mentioned are not new and the fact that some of these methods are nondestructive is doubtful. First of all, there should be a clarification of what destructive and nondestructive method means, and the methods mentioned should be analyzed by starting from these definitions.
More observations can be found directly on the text.

Author Response
The topic of this paper is still relevant. Literature in the field presents numerous studies in which various tree testing methods have been investigated with the purpose of determining tree stability and the risk that trees pose. However, the article does not have a high degree of novelty and originality. There are numerous articles that have approached this topic over the past 30 years. The methods mentioned are not new and the fact that some of these methods are nondestructive is doubtful. First of all, there should be a clarification of what destructive and nondestructive method means, and the methods mentioned should be analyzed by starting from these definitions.
Response: This is a review article summarizing the nondestructive testing methods that are developed for inspecting urban trees for public safe with a focus on the techniques that are most fit to field inspection practices. A "Discussion and Conclusion" section has been added to clarify the commonly used term "nondestructive testing" with respect to practical applications.
Reviewer 2 Report
The paper very briefly desribed the methods (you can make for each method and results a single or more papers). The literature is adequate and satisfactory.
A comment:
- line 54: delete word density.
Author Response
A comment:
- line 54: delete word density.
Response: Correction has been made.
Reviewer 3 Report
Generally the methods are well described. for better clarity, I suggest adding pictures, resp. pictorial procedures of individual methods. This will increase the level of the article. There is no summary/conclusion and possible discussion that would summarize the advantages and disadvantages of each method. I also think that the literary research could have been enriched by a few more works.
In forestry, the method of digital image (eg photography) and its subsequent analysis by specialized software (eg ImageJ) for the evaluation of external qualitative features are also used. Please add these methods as you see fit.
Formal suggestions:
Terminology...i think the right term for the "failure"/""defect" is "negative qualitative feature" (according to EN 1309-3).
Please check the sentence on line nr. 54.
Author Response
Generally the methods are well described. for better clarity, I suggest adding pictures, resp. pictorial procedures of individual methods. This will increase the level of the article.
Response: Four figures have been added to illustrate the concept and use of several commonly used tree inspection methods.
There is no summary/conclusion and possible discussion that would summarize the advantages and disadvantages of each method. I also think that the literary research could have been enriched by a few more works.
In forestry, the method of digital image (eg photography) and its subsequent analysis by specialized software (eg ImageJ) for the evaluation of external qualitative features are also used. Please add these methods as you see fit.
Response: A "Discussion and Conclusion" section has been added. A new section on "Electric Resistivity Tomography and Infra-Red Thermography" have been added.
Formal suggestions:
Terminology...i think the right term for the "failure"/""defect" is "negative qualitative feature" (according to EN 1309-3).
Please check the sentence on line nr. 54.
Response: checked and correction has been made.
Round 2
Reviewer 1 Report
The review does not have a high degree of novelty and originality. There are numerous articles that have approached this topic over the past 30 years. The methods mentioned are not new and the fact that some of these methods are non-destructive is doubtful.
I continue to believe that ”remote sensing” is not a method for the non-destructive analysis of trees. It rather helps identify trees that may present a risk due to their being too close to buildings, playgrounds, parking lots, roads and railways as well as trees with a tilted stem or an asymmetrical crown, etc. Later on, trees identified this way can be analysed by using the methods described in the paper.
I appreciate the author’s effort to include electric resistivity tomography and infra-red thermography in the paper.
Regarding the Discussions and Conclusions section I have the following observations:
- There is a discrepancy between the title and this paragraph (lines 207 – 208). The title is about the methods used in tree analysis and not about the analysis of wood as a material. In the case of trees, apart from wood analysis, the term non-destructive must also take into consideration the subsequent reaction of trees after performing the analysis, the probability that they might be infested with xylophagous fungi, bacteria or viruses, etc. and thus, the probability that they might develop abnormalities or defects that could endanger their stability. The papers cited in this paragraph refer to wood products, wood structures and not at all to living trees.
- By analysing these lines ( lines 209 – 213) there is one question that comes to my mind: How is the analysis using Pressler’s borer more destructive?
- The motivation that arborists use this methods widely (lines 214 – 216) is not convincing. Arborists are not scientists or researchers (in most cases). If the author chooses to insist on this, maybe the title of the article should be changed to – Frequent / Common methods for testing urban trees – eliminating, thus, the word ”non-destructive”. In this case, the text should be adapted too.